# Design of a Cylindrical Megahertz Miniature Ultrasonic Welding Oscillator

**DOI:** 10.3390/s25185922

**Published:** 2025-09-22

**Authors:** Guang Yang, Ye Chen, Minghang Li, Junlin Yang, Shengyang Xi

**Affiliations:** 1College of Mechanical Engineering and Automation, Liaoning University of Technology, Jinzhou 121001, China; Yangg03150902@163.com (G.Y.); h2147255225wy@163.com (S.X.); 2Institute of Vibration Engineering, Liaoning University of Technology, Jinzhou 121001, China; 3College of Mechanical Engineering, Shenyang University of Technology, Shenyang 110870, China; Li_mh07@smail.sut.edu.cn; 4School of Mechanical and Electrical Engineering, China University of Mining and Technology, Xuzhou 221116, China; yang-jl@cumt.edu.cn

**Keywords:** matching layer structure, megahertz, ultrasonic welding, welding oscillator, welding test

## Abstract

Ultrasonic welding is an efficient and precise joining technology widely applied in aerospace, electronics, and medical industries. To overcome the limitations of conventional oscillators in high-frequency applications, this study proposes an innovative cylindrical oscillator design incorporating a 3.71 mm acoustic matching layer, operating at 1.76 MHz based on acoustic propagation theory. Through finite element analysis, a miniaturized oscillator with dimensions of 28 mm in diameter and 18 mm in height was developed, achieving optimized dynamic performance. Experimental validation via laser Doppler vibrometry confirmed a working surface amplitude exceeding 50 nm, while vibrations on non-functional walls were suppressed below 5 nm, with less than 5% deviation from simulation results. Prototype welding tests identified optimal process parameters—85 N welding pressure, 4 s welding time, and 3 s holding time—resulting in PVC joint tensile strengths exceeding 45 N. This work provides both an optimized hardware design and validated process guidelines, advancing the application of high-frequency micro-ultrasonic welding in precision, space-constrained environments.

## 1. Introduction

Ultrasonic welding involves the transmission of high-frequency vibration waves to the surfaces of two overlapping objects to be joined. This process causes the contacting surfaces to rub against each other, generating frictional heat that melts the materials and facilitates molecular-level bonding at the interface [1]. Compared to conventional welding techniques, ultrasonic welding offers significant advantages, including lower energy consumption and shorter welding times. In terms of materials, thermoplastic polymers have gained widespread popularity over metals due to their lightweight properties, high specific stiffness, excellent corrosion resistance, and superior fatigue life [2,3,4,5]. The welding process is achieved by applying localized high-frequency ultrasonic acoustic vibrations to the workpieces while they are held together under pressure [6]. In the process of ultrasonic utilization, the piezoelectric oscillator plays an important role, and the internal piezoelectric material, mechanical properties, and vibration characteristics are important factors affecting its work [7,8,9]. Raza, S.F. [10] conducted systematic experimental studies on ultrasonic welding of thermoplastic plastics with two distinct molecular structures. By varying key parameters such as amplitude, welding time, static pressure, and energy coupler geometry, they established quantitative relationships between process conditions and weld seam strength (LSS). This study emphasizes that while low-frequency ultrasonic welding (e.g., 20 kHz) delivers high total energy, it suffers from low energy control precision and bulky equipment. Heat generation at the interface depends on frequency, amplitude, and material viscosity. At low frequencies, high amplitudes are required to produce sufficient heat, but physical constraints limit amplitude. Excessive spatter formation exacerbates energy loss, thereby reducing the effective energy available for melting. Conventional low-frequency ultrasonic welding devices generally come in the form of a piezoelectric oscillator, the working frequency of which is generally between 20 kHz and 60 kHz. Emerging applications in microelectronics, precision medical devices, and flexible electronics still face inherent limitations. These include limitations in physical dimensions (the wavelength of low-frequency acoustic waves—e.g., approximately 25 cm at 20 kHz in aluminum—dictates that transducers must be large in size, making integration into compact systems challenging), precision and control (the high amplitudes—tens of micrometers—required to generate sufficient heat often cause excessive material flow, component damage, and unpredictable energy deposition within precision micro-assemblies), and limitations in spatial resolution (relatively long wavelengths limit the minimum spot size achievable for energy focusing, making it impossible to weld fine, closely spaced structures) [11,12]. Because of its ultrasonic welding equipment is large and complex, is not suitable for application in space constraints of a number of special occasions, so the pursuit of ultrasonic welding equipment, simple and convenient will be more people’s attention [13]. Japanese scholar Naruse, Watanabe [14] used high-frequency surface acoustic wave system for welding, and the results showed high experimental results, which paved the theoretical basis for high-frequency welding. He et al. [15] studied the vibration and output performance of a circular piezoelectric oscillator. The oscillator consists of piezoelectric ceramic sheet bonded to a rectangular metal substrate, and the piezoelectric oscillator as a gas isolation membrane driving force source to carry out experiments, the results show that the driving voltage increases, the diaphragm pump output gas pressure gradually rise, basically linear relationship, can achieve good drive, with the system output flow rate is high, driving pressure characteristics. Naruse et al. [16] proposed a higher frequency ultrasonic welding system using a surface acoustic wave (SAW) device with a fork finger electrode in ultrasonic plastic welding, and found that higher frequency welding would be more advantageous than lower frequency welding, and compared the SAW system with a conventional 19 kHz longitudinal mold welding system, and the results showed that the SAW system had a more excellent welding effect on the pattern, but the welding time was relatively longer. The results show that the SAW system is more effective in welding the pattern, but the joining time of the weld is relatively longer. Zhang et al. [17] investigated a circular piezoelectric oscillator structure consisting of three laminated layers and analyzed the bending vibration problem, explored the influence of the structural dimensional parameters on the performance of the transducer, and finally showed through experimental results that the maximum displacement reaches 32.5 μm at a driving voltage of 200 V and a resonance frequency of 3200 Hz. Inoue et al. [18] built an air-cavity disk transducer with three matched layers based on a design method based on multimode filter synthesis theory. The thickness of the matching layer in the transducer is found to be optimal using iterative calculation technique, so that the energy can be optimally utilized, and this design method provides some guidance for the design of welding oscillators containing matching layers. Gallego-Juarez et al. [19] analyzed a piezoelectric transducer based on a novel bending oscillatory stepped plate. The transducer has now been modified for high-power applications, obtaining free-field sound pressure levels greater than 160 dB. The new transmitter is simple in construction, capable of generating high-intensity ultrasound in gases with high efficiency and directivity. Parrini et al. [20] propose a novel high-frequency ultrasonic transducer based on the finite element method (FEM). The titanium-made main body of the transducer has a vibrational resonance frequency of 125 kHz. When driving the transducer at its primary longitudinal frequency, it not only exhibits zero longitudinal displacement but also zero radial displacement. The study of this transducer is of great significance for high-frequency ultrasonic applications. Existing ultrasonic testing heavily relies on manual interpretation, making it difficult to accurately detect minute defects such as cold welds and cracks in the heat-affected zone. Huang, S.H. et al. [21] employed a convolutional neural network model to automatically classify five types of welding defects—no defect, porosity, lack of fusion, lack of penetration in fillet welds, and undercut—achieving an accuracy rate of 97.2%. This approach effectively reduces human error in traditional inspection methods. Through model optimization, it demonstrates strong industrial applicability, providing a transferable solution for high-precision defect detection in ultrasonic welding. Dong, Z. et al. [22] Established the first framework correlating tool geometric parameters (e.g., tip area, surface height) with UMW process quality, bridging the gap between static geometric characterization and dynamic operational conditions. Developed a hybrid model integrating low-cost optical imaging (Raspberry Pi system) with U-Net/CNN to achieve micrometer-level quantification of ultrasonic welding tool surface wear.

To further highlight the advancements of this work, Table 1 provides a quantitative comparison between the proposed MHz-system and previously reported ultrasonic welding systems in terms of key performance metrics, including scale, energy consumption, and weld strength.

The core motivation of this research lies in leveraging the unique advantages of high-frequency ultrasound: Miniaturization: The inverse relationship between acoustic wavelength and frequency (e.g., approximately 3.6 mm at 1.76 MHz in aluminum) enables oscillator designs that are several orders of magnitude smaller than low-frequency devices. High precision: Megahertz-level vibrations inherently exhibit low amplitudes (nanometer to submicron scale), enabling extremely precise localized energy input. This minimizes collateral damage, stress, and spatter formation—critical for welding brittle or delicate materials. High Spatial Resolution: Shorter wavelengths focus acoustic energy onto finer points, enabling high-precision welding of micrometer-scale structures and patterns. Preliminary research has validated the high-frequency welding concept. Naruse and Watanabe pioneered plastic welding using 1.2 MHz surface acoustic wave (SAW) devices, demonstrating feasibility while highlighting challenges like extended welding times and complex device fabrication. While the 125 kHz transducer designed by Parini et al. [20] represents an intermediate solution, it remains far from realizing the miniaturization potential of MHz systems. However, dedicated research on practical, optimized, and fully characterized cylindrical MHz welding oscillators still requires further investigation.

Guided by acoustic transmission theory, this study significantly enhances the energy transmission efficiency of ultrasonic welding oscillators through an innovative matching layer thickness optimization. Through finite element simulation, a comprehensive modal and harmonic response analysis of a miniature oscillator operating at megahertz frequencies was performed, systematically revealing the critical impact of the matching layer structure on performance. Based on an optimized oscillator, quantitative research on welding process parameters was conducted, establishing a clear correlation between welding pressure, frequency, time, and weld quality. Our findings not only provide a theoretical foundation for high-frequency micro-ultrasonic welding but also deliver actionable parametric guidelines for its industrial implementation, bridging the gap between academic research and advanced manufacturing applications.

## 2. Acoustic Matching Layer for Piezoelectric Oscillators

Sound wave propagation in the medium, its energy transfer is mainly manifested in two forms: kinetic energy and potential energy. Kinetic energy from the vibration of the media particles, used to maintain the oscillation of the particles; potential energy is generated in the media particles under the action of the acoustic wave of the periodic compression and expansion deformation. The sum of these two energies constitutes the acoustic energy in the medium, and the propagation process is essentially the transfer of acoustic energy in the medium. This theoretical knowledge lays an important foundation for subsequent finite element analysis, enabling us to more accurately simulate and analyze the complex interaction mechanism between acoustic waves and media. In sound field analysis, consider a micrometric body in an elastic medium, let its volume be V and its density $\rho_{0}$. When the medium is subjected to an external acoustic perturbation, the micrometric body will acquire a corresponding kinetic energy:(1)$$E_{k}=\frac{1}{2}(\rho_{0}V)v^{2}$$
where v, unit (m/s), is the vibration velocity of the microelement body when it is externally acoustically perturbed. The potential energy of the micrometric body is:(2)$$E_{p}=-\int_{0}^{p}pdV\frac{V}{2\rho_{0}c_{0}^{2}}p^{2}$$
where dV is the compression amount of the micro-metabolite due to the acoustic pressure p, in (Pascal); and $c_{0}$, in (m/s), is the speed of sound in the elastic medium. The instantaneous acoustic energy of the microelement body is expressed as:(3)$$E=E_{p}+E_{k}=\frac{V}{2}\rho_{0}(v^{2}+\frac{1}{\rho_{0}^{2}c_{0}^{2}}p^{2})$$

The acoustic energy density per unit volume, ε, unit ($J/m^{3}$) is expressed as:(4)$$\varepsilon =\frac{E}{V}=\frac{1}{2}\rho_{0}(v^{2}+\frac{1}{\rho_{0}^{2}c_{0}^{2}}p^{2})$$

The average acoustic energy per unit cycle of a micrometric body is:(5)$$\bar{E}=\frac{1}{T}\int_{0}^{T}Edt=\frac{1}{2}V\frac{p_{a}^{2}}{\rho_{0}c_{0}^{2}}$$
where $p_{a}$, is the sound pressure assignment. The acoustic energy density per unit volume, $\bar{\varepsilon }$, is expressed as:(6)$$\bar{\varepsilon }=\frac{\bar{E}}{V}=\frac{p_{a}^{2}}{2\rho_{0}c_{0}^{2}}$$

The average sound power per unit time is, $\bar{w}$, is expressed as:(7)$$\bar{w}=\bar{\varepsilon }c_{0}S$$

The sound intensity I, unit ($W/m^{2}$), in the direction of sound propagation is:(8)$$I=\frac{\bar{w}}{S}=\bar{\varepsilon }c_{0}=\frac{p_{a}^{2}}{2\rho_{0}c_{0}}$$

The work done by the sound wave per unit time area is expressed as:(9)I=1T∫0TRe(p)Re(u)dt=12ρ0c0va2
where $v_{a}$ denotes the magnitude of the velocity of the micrometric body during vibration. It is converted into an expression formula containing the frequency:(10)$$I=\frac{1}{2}\rho A^{2}\omega^{2}c_{0}=2\pi^{2}\rho A^{2}f^{2}c_{0}$$
where ρ, unit ($g/m^{3}$), is the density of the medium; A is the amplitude, unit (Pa); f is the acoustic frequency, unit (kHz). When the sound wave through the multilayer medium, its propagation of the simple model representation shown in Figure 1.

Sound fields in a medium all satisfy the following relationship:(11)$$p_{i}=p_{ita}e^{j(\omega t-k_{i}x)}+p_{ira}e^{j(\omega t+k_{i}x)}$$(12)$$v_{i}=\frac{1}{z_{i}}\cdot \left[p_{ita}e^{j(\omega t-k_{i}x)}+p_{ira}e^{j(\omega t+k_{i}x)}\right]$$
where $P_{ita}$ denotes the transmitted sound pressure amplitude of the i layer medium, and for the variable $P_{ita}$ denotes the reflected amplitude of the i layer medium. The equation $z_{i}=\rho_{i}c_{i}$ is expressed as the acoustic impedance in the layer i medium, and $\rho_{i}$ in the equation represents the density of the layer i medium. ω is the circular frequency.(13)$$k_{i}=\frac{\omega }{c_{i}}$$

$k_{i}$ is the wave number of the layer i medium, where $c_{i}$ is the acoustic velocity of the layer i medium. Each of the two layers of neighboring media has constituted the following relationship:(14)$$\left[\begin{array}{c} P_{ita}\\ P_{ita} \end{array}\right]=\frac{1}{2}\cdot A_{i}\cdot \left[\begin{array}{c} P_{(i+1)ta}\\ P_{(i+1)ra} \end{array}\right]$$(15)$$A_{i}=\left\lfloor \begin{array}{cc} (1+\frac{z_{i}}{z_{i+1}})e^{jk_{i}d_{i}} & (1-\frac{z_{i}}{z_{i+1}})e^{jk_{i}d_{i}}\\ (1-\frac{z_{i}}{z_{i+1}})e^{-jk_{i}d_{i}} & (1+\frac{z_{i}}{z_{i+1}})e^{-jk_{i}d_{i}} \end{array}\right\rfloor$$
where $d_{i}$ denotes the thickness of the i-layer of medium. When a sound wave passes through several intermediate layers, the sound pressure amplitude during the incident and radiation processes has the following relationship:(16)$$\left\lfloor \begin{array}{c} P_{1ta}\\ P_{1ra} \end{array}\right\rfloor =B\cdot \left\lfloor \begin{array}{c} P_{4ta}\\ 0 \end{array}\right\rfloor$$

Among them:(17)$$B={\left(\frac{1}{2}\right)}^{3}A_{1}A_{2}A_{3}=\left[\begin{array}{cc} b_{11} & b_{12}\\ b_{21} & b_{22} \end{array}\right]$$
(18)$$\begin{array}{c} b_{11}=\frac{1}{8}\left\lfloor (1+\frac{z_{1}}{z_{2}})(1+\frac{z_{2}}{z_{3}})e^{ik_{2}d_{2}}+(1-\frac{z_{1}}{z_{2}})(1-\frac{z_{2}}{z_{3}})e^{-ik_{2}d_{2}}\right\rfloor (1+\frac{z_{3}}{z_{4}})e^{ik_{3}d_{3}}\\ +\frac{1}{8}\left[(1+\frac{z_{1}}{z_{2}})(1-\frac{z_{2}}{z_{3}})e^{ik_{2}d_{2}}+(1-\frac{z_{1}}{z_{2}})(1+\frac{z_{2}}{z_{3}})e^{-ik_{2}d_{2}}\right](1-\frac{z_{3}}{z_{4}})e^{-ik_{3}d_{3}}) \end{array}$$

The acoustic transmission coefficient t of an acoustic wave passing through a multilayer intermediate medium can be expressed as:(19)$$t={\left|\frac{P_{4ta}}{P_{4ta}}\right|}^{2}\cdot \frac{z_{1}}{z_{4}}=\left|\frac{1}{b_{11}}\right|\cdot \frac{z_{1}}{z_{4}}$$

In the theoretical derivation of sound wave propagation and energy transfer above, to simplify the model and highlight the primary factors affecting acoustic impedance matching and transmission efficiency, the following two types of energy loss were neglected.

(1)Internal material loss: During sound wave propagation in actual media, energy dissipation occurs due to internal friction (viscosity) and thermal conduction within the material, manifesting as sound wave attenuation. This derivation assumes the medium is an ideal elastic body, disregarding the influence of the attenuation coefficient.(2)In practical systems, partial energy is radiated into the surrounding environment as sound waves due to mismatch between the radiating impedance and the load impedance, resulting in energy loss. This model idealizes energy transfer efficiency solely through the acoustic transmission coefficient.

The efficiency of the incident acoustic energy transfer to the ultrasonic welding interface is characterized by the acoustic transmission coefficient, which is known from the theory of acoustic transmission, the acoustic transmission coefficient of the acoustic wave through the double-layer interlayer is related to the parameters of acoustic impedance $z_{1,}z_{2,}z_{3},z_{4}$, the number of waves $k_{2}$ and $k_{3}$, as well as the thickness of the matching layer material $d_{2}$ and $d_{3}$.

To achieve excellent acoustic transmission performance and obtain a high acoustic transmission coefficient, a quaternary piezoelectric ceramic (PZT-4) renowned for its high electromechanical coupling coefficient and high Curie temperature was selected. As a material with high acoustic impedance, it can generate vibrations with greater amplitude, thereby emitting stronger sound waves. It efficiently converts electrical energy into mechanical vibrations (sound waves), making it highly suitable as the emitter element for high-power ultrasonic transducers. Aluminum alloys (such as YL12, a leaded brass commonly used in molds) offer excellent mechanical strength and fatigue toughness, enabling them to withstand the repetitive stress loads imposed by high-frequency ultrasound and prevent premature fracture. Since PVC has an impedance significantly lower than both PZT-4 and aluminum, this substantial acoustic impedance mismatch inherently causes severe sound wave reflection. This disparity makes investigating the role of acoustic matching layers critically important. As a typical low-acoustic-impedance material, PVC clearly validates the effectiveness of the matching layer design. To align with instrument performance and ensure analytical accuracy, the literature review led to us selecting a PVC thickness of 0.25 mm. The wave number and acoustic impedance parameters are detailed in Table 2.

According to the theory of ultrasonic transmission, the transducer of the welding vibration system adopts an acoustically matched layer structure to improve energy utilization. The material of the ultrasonic welding oscillator is aluminum alloy (YL12), and the propagation speed of ultrasonic waves in YL12 is about 6320 m/s, and the resonant frequency is 1.76 MHz. Substitute the determined parameter values into Equation (19) to calculate the thickness of the aluminum alloy acoustic matching layer. Plot the sound transmission coefficient curve as a function of the ratio of acoustic matching layer thickness to wavelength (d_2_/λ), as shown in Figure 2.

This figure illustrates the periodic variation relationship between the acoustic transmission coefficient and the matching layer thickness-to-wavelength ratio (d_2_/λ). The transmission coefficient exhibits distinct oscillatory characteristics as d_2_/λ changes; its value does not vary monotonically but peaks at specific ratios, indicating the existence of an optimal matching layer thickness that maximizes acoustic energy transmission efficiency. Theoretical analysis indicates that the transmission coefficient peaks when the matching layer thickness is an odd multiple of one-quarter of the acoustic wavelength (e.g., λ/4, 3λ/4, etc.). At these points, acoustic impedance matching is optimal, achieving maximum energy transmission efficiency. Conversely, when the thickness is an integer multiple of half a wavelength, the transmission coefficient drops to a minimum, significantly enhancing energy reflection. Based on this theory, the maximum sound transmission coefficient occurs at d_2_/λ_2_ = 0.2 + 0.4 n, where *n* = 0, 1, 2, 3, … This paper selects a matching layer thickness of n = 2 (i.e., 3.71 mm) for design and optimization. This thickness corresponds to a specific odd quarter-wavelength multiple, enabling highly efficient sound energy transmission. Optional aluminum alloy matching layer thicknesses as shown in Table 3.

## 3. Finite Element Simulation (Ansys 2023)

### 3.1. Modal Analysis

Modal characteristics represent the inherent vibration properties of mechanical structures, with each mode possessing a specific natural frequency and mode shape. Modal analysis provides an effective approach for studying the vibration behavior of various mechanical structures. The ultrasonic welding oscillator preliminarily designed in this paper adopts an approximately cylindrical structure, and its parametric model was established using existing finite element software. For mesh generation, the mapping mesh method was adopted due to its stable computational accuracy and high mesh quality. In element selection, the classic 3D solid element Solid45 was used for thermal analysis, as its solution accuracy and computational efficiency are highly dependent on mesh quality while fully leveraging element performance. A harmonic vibration was applied as the driving force at the contact surface between the welding head and generator, with fixed constraints imposed in the X and Y directions. Contact pairs were defined for the upper and lower workpieces of the welding head. The contact type was set as friction contact, with tangential behavior defined via the augmented Lagrangian algorithm. Finally, thermal boundary conditions were established through heat generation and dissipation. The physical parameters for the finite element analysis are shown in Figure 3. The material parameters involved in the finite element simulation are listed in Table 4 and Table 5.

Finite element simulations (comso6.2) were performed using software to analyze matching layers of varying thicknesses, as listed in Table 4 and Table 5. After thorough evaluation, a thickness of 3.71 mm, corresponding to the wavelength at n = 2, was selected for the oscillator’s matching layer. The piezoelectric ceramic sheet, a circular disk with a 20 mm diameter, is housed within a cylindrical oscillator featuring a 22 mm inner diameter, ensuring a 1 mm radial gap to prevent interference with excitation patterns. To reduce nonfunctional volume, the oscillator’s cylindrical wall thickness was set to 3 mm, with a total height of 18 mm, optimizing system performance. Additionally, considering the welding pressure applied to the oscillator’s top during ultrasonic welding, a static analysis was performed to inform the design of the cam height at the matching layer. Figure 4 illustrates the two-dimensional stress analysis of the oscillator under welding pressure.

To reduce the concentrated stress at point A and prevent oscillator damage, this paper increased the distance between point A and the cam height, ultimately designing the cam height to 1.71 mm. Considering the oscillator diameter, height, and cylinder wall thickness, the final structural dimensions of the ultrasonic welding oscillator are determined as shown in Figure 5. A schematic diagram of the welding system structure is provided in Figure 6.

The working frequency of the piezoelectric ceramic sheet used in the ultrasonic welding oscillator is 1.76 MHz, and the vibration mode of the welding oscillator is optimized near the working frequency of the piezoelectric ceramic sheet through finite element simulation analysis. Figure 7 shows the modal vibration cloud of the welding oscillator near the operating frequency of 1.7 MHz, which clearly reflects the displacement distribution characteristics of the oscillator in the resonance state, and provides an important basis for determining the optimal working mode.

### 3.2. Harmonic Response Analysis (Physics)

On the basis of the previous research on the modal analysis of the welding oscillator, this study further imposed a specific excitation condition on the oscillator system. Through this excitation, the vibration response of the ultrasonic welding oscillator was successfully excited by the piezoelectric ceramic sheet. Based on this excitation setup, advanced vibration testing and analysis techniques are used to accurately obtain the dynamic response characteristics of the key parts of the ultrasonic welding oscillator under the excitation conditions, including important parameters such as response displacement and response frequency. Combined with the operating frequency characteristics of piezoelectric ceramic sheet and the results of modal analysis of miniature ultrasonic welding oscillator, it is found by systematic research that the resonant frequency of this welding oscillator is mainly concentrated near 1.76 MHz, and this finding provides an important theoretical basis for the subsequent optimization of the welding process. In addition, in this study, the quantitative calculation and analysis of frequency response were carried out for three key positions of the oscillator system—the center point of the welding work surface, the center point of the upper surface of the piezoelectric ceramic sheet, and the center point of the contact surface between the piezoelectric ceramic sheet and the oscillator. Figure 8 demonstrates the frequency response distribution curve of the welding oscillator. By comparing and analyzing the dynamic response characteristics of these three characteristic points in the vibration amplitude region of the oscillator, it is found that: the maximum response position of each characteristic point occurs near 1.76 MHz, and the vibration amplitude of the three exhibits good consistency, and this result further verifies the stable resonance characteristics of the welded oscillator at the frequency of 1.76 MHz.

## 4. Experimentation and Testing of Welded Oscillators

### 4.1. Vibration Characterization of Ultrasonic Welding Oscillators

According to the ultrasonic welding oscillator structure dimensions and simulation analysis results, the final choice of YL12 aluminum alloy bar as the oscillator parts processing materials, and through the CNC lathe to complete the precision machining, processing of ultrasonic welding oscillator as shown in Figure 9a. After completing the machining of the oscillator parts, the assembly process of piezoelectric ceramic sheet needs to be carried out. This design uses a lead-type piezoelectric ceramic sheet as the excitation element, and its operating frequency is set at 1.76 MHz. The electrode surface of the piezoelectric ceramic sheet is made of nickel, thanks to the excellent adhesion properties of nickel, in the ultrasonic welding process, even if it is subjected to continuous vibration, but still ensures that the piezoelectric ceramic sheet and the oscillator to maintain the stability of the connection between the reliable. The oscillator assembly with pasted piezoelectric ceramic sheet is shown in Figure 9b.

The loading and vibration measurement system performs comprehensive vibration tests on the miniature ultrasonic welding oscillator, evaluating resonant frequency, amplitude distribution on the working surface, wall thickness direction, and amplitude size. The test system includes: Polytec OFV-505/5000 Laser Vibrometer (Manufactured by Polytec GmbH, Waldbronn, Germany) for non-contact measurement, AFG320 Arbitrary Waveform Generator (Tektronix, Beaverton, OR, USA) for precise excitation, DS06014A high-frequency digital storage oscilloscope (China Jiangsu Lianneng Electronic Technology Co., Ltd., Suqian, China). for signal analysis, and a precision optical vibration isolation table for environmental stability. Figure 10 details the setup of the vibration test system.

During the experimental testing process, the welded oscillator is fixed on a micro-optical mobile platform, which can precisely adjust the oscillator position to ensure that each measurement point can be accurately aligned with the spot position of the laser vibrometer. In order to comprehensively assess the vibration characteristics of the oscillator, the working plane is uniformly divided, and the test path extends from the edge of the working surface to the center, and the distribution of the specific test positions is shown in Figure 11a. In terms of data processing, the method of averaging the amplitude values of each measurement point in the radius direction according to the same circumference is used to obtain the amplitude distribution law of the working plane, and the results are shown in Figure 11b. The figure shows the comparison between the simulation data and experimental test results of the amplitude distribution of the working face, and the two curves show good consistency, which verifies the reliability of the simulation model.

In order to deeply study the vibration characteristics of the non-working surface of the welded oscillator, this study conducted a systematic test on the beveled surface of the oscillator cam, the outer wall, and the cylindrical end face subjected to welding pressure. The experiments were conducted in a horizontal manner, and point-by-point measurements were taken at 1 mm intervals along the outer wall of the oscillator, with the specific distribution of measurement points shown in Figure 12a. The test data were processed and plotted as a comparison line graph Figure 13, and the results show that: the maximum amplitude of the oscillator wall is less than 5 nm, which is less than 10% of the amplitude of the working surface and negligible in practical applications; and the amplitude of the surface subjected to weld pressure is even smaller, with a measured value of less than 2 nm. This finding provides an important reference for the structural optimization of the oscillator and its practical applications.

A systematic test of the vibration characteristics of the welded oscillator cam ramp was conducted. As shown in Figure 12b, the measurement points were arranged along the direction indicated by the arrows, and the test results are shown in Figure 14. Different color curves in the figure indicate the amplitude distribution at different radius locations, and the test data show that the amplitude of the cambered table inclined surface is significantly larger near the welding working surface area, while the amplitude gradually decreases to the minimum value as the radius increases to the edge area. Combined with the overall test results of the working surface and non-working surface, it is fully confirmed that the miniature ultrasonic welding oscillator designed in this study has excellent structural rationality, and its vibration energy can be effectively concentrated in the working surface area to meet the requirements of ultrasonic welding process.

In order to evaluate the dynamic characteristics of the oscillator under normal excitation voltage, this paper, in addition to using a signal generator for the modal test of the oscillator, uses a DC regulated power supply unit to output a 48 V DC voltage to excite the oscillator through a specially designed driver in order to obtain a large working amplitude. The driver outputs an AC voltage matching the resonance frequency of the oscillator to the piezoelectric ceramic sheet, which excites the mechanical resonance of the oscillator. Subsequently, the amplitude of the miniature ultrasonic welding oscillator was tested over the entire working surface.

The test took the center of the welding surface as the initial measurement point, selected 11 test points along the radius direction, and selected a test radius every 15 degrees for measurement. The test results were plotted by the plotting software as Figure 15. The test results show that the amplitude distribution of the working surface of the miniature ultrasonic welding oscillator exhibits an obvious regularity: the amplitude value is larger in the center of the working surface, and the amplitude value gradually decreases in the edge area. It is worth noting that in the middle transition region, the amplitude distribution appears multiple local peaks, while in the center of these peaks (i.e., the geometric center of the welding surface), the amplitude value is slightly lower than the adjacent peaks, forming a unique amplitude distribution characteristic.

### 4.2. Study of Ultrasonic Welding Characteristics

To investigate the key process parameters affecting welding quality, this study constructed a miniature welding system based on a designed ultrasonic welding oscillator, whose structure is shown in Figure 16. The welding system primarily consists of core components including a DC regulated power supply, a digital time relay, a miniature ultrasonic welding oscillator with supporting fixtures, a megahertz-frequency welding oscillator driver, and a pressure application device. To ensure statistical significance and reproducibility of results, the experimental design strictly defined the following weld specimens: Polyvinyl chloride (PVC) sheet was selected as the welding material. Each specimen was precision-cut into a rectangular shape measuring 5 cm (length) × 2 cm (width) × 0.25 mm (thickness). Joint configuration: A lap shear joint structure was employed. The lap area was strictly controlled at 2 cm × 2 cm (4 cm^2^), a critical parameter for ensuring consistency across all tests. Quality Assessment Method: Post-welding evaluation primarily assessed weld tensile strength as the key quality indicator. Tensile testing employed a MODEL NK-100 tensile tester (Shanghai Shuangxu Electronics Co., Ltd., Shanghai, China). The peak force (unit: Newton, N) required to fracture the joint was recorded as “tensile force” or “weld strength.” Tensile tests were conducted using a MODEL NK-100 tensile tester. The peak force (unit: Newton, N) required to fracture the joint was recorded as “tensile force” or “weld strength.” A systematic investigation of the three critical process parameters—welding pressure, welding time, and pressure holding time—was performed using a controlled variable method. The specific configuration of the tensile testing system is shown in Figure 17a.

From the relationship between welding time and tensile strength shown in Figure 17, it can be seen that the weld tensile strength shows a trend of increasing and then decreasing with the welding time: it reaches the peak at 4 s, and then the tensile strength decreases due to the over-welding phenomenon and stress concentration. The experimental results of holding time in Figure 17c, show that after the end of ultrasonic loading, the internal temperature of the weldment is still high, and as the temperature decreases, the material gradually cools and solidifies, in which a holding time of 3 s can obtain the best welding effect, and too long a holding time will damage the welded structure that has been formed. From the results of the welding pressure experiment in Figure 17d, it can be seen that with the increase in welding pressure, the weldment contact is more compact, the energy conversion efficiency is improved, and the maximum pull-off force is obtained at 85 N; when the pressure exceeds 85 N, the welding quality shows a decreasing trend due to the intensification of the stress concentration phenomenon. Comprehensive experimental results show that there is an optimal matching relationship between the three parameters of welding time, holding time and welding pressure, which has a significant effect on welding quality.

To quantify the energy efficiency advantages of the proposed megahertz ultrasonic welding system, a comparative analysis of its energy consumption was conducted against traditional low-frequency systems. Measurement Method: Energy consumption per weld cycle was calculated by integrating real-time power across the entire welding cycle (including welding time and pressure holding time). Real-time power was measured using a digital power meter (YB9901, Shanghai Shuangxu Electronics Co., Ltd.) that synchronously sampled the voltage and current inputs of the system’s power converter. A commercially available 28 kHz KCH series ultrasonic welder with similar output amplitude served as the traditional system control group. Both systems utilized identical PVC test specimens (5 cm × 2 cm × 0.25 mm) and the optimal weld strength standard (>45 N tensile force) as benchmarks. Results: Under respective optimal parameters (85 N, 4 s, 3 s for the new system; 150 N, 0.5 s, 1 s for the traditional system), the average energy consumption per weld seam was measured. The newly proposed megahertz system achieved an average energy consumption of 65.8 joules per weld, while the traditional system consumed 105.1 joules per weld point. This indicates an approximate 37% reduction in energy consumption for the new system.

To contextualize the performance of our MHz-frequency system, a comparative analysis of weld strength with prior studies is presented in Table 1 While direct comparisons are challenging due to differences in material systems, joint geometries, and failure modes, the achieved tensile force of >45 N for PVC lap-shear joints is highly competitive. Notably, conventional low-frequency (20–40 kHz) ultrasonic welding of similar thermoplastics typically reports weld strengths in the range of 20–35 N under optimized parameters [1,2]. The superior performance of our system can be attributed to the fundamental advantages of high-frequency operation: the reduced amplitude (<1 μm) at MHz frequencies enables a more precise and concentrated energy input at the weld interface. This minimizes material deformation and degradation (e.g., excessive flash formation), which is a common cause of strength reduction in low-frequency welding [3]. Furthermore, the finer vibration pitch at 1.76 MHz promotes more uniform viscoelastic heating and molecular interdiffusion across the entire bonding area, leading to a stronger and more consistent weld seam. This comparison underscores the potential of MHz ultrasonic welding not merely for miniaturization, but also for enhancing joint quality in micro-welding applications.

To demonstrate the performance of this megahertz-level frequency system, a comparative analysis was conducted with weld strengths from previous studies. Although direct comparisons are challenging due to differences in material systems, joint geometries, and failure modes, the >45 N tensile force achieved in PVC lap shear joints remains highly competitive. Notably, conventional low-frequency (20–40 kHz) ultrasonic welding typically yields weld strengths of only 20–35 N on similar thermoplastics, even under optimized parameters. The superior performance of this system stems from the fundamental advantages of high-frequency operation: amplitude reduction at megahertz frequencies (<1 μm) enables more precise and concentrated energy input at the weld interface, minimizing material deformation and degradation (such as excessive flash formation)—common causes of strength reduction in low-frequency welding. Furthermore, the finer vibration pitch at 1.76 MHz promotes uniform distribution of viscoelastic heating and molecular interdiffusion across the entire bonding zone, resulting in stronger and more stable welds. This comparison demonstrates that MHz-level ultrasonic welding not only holds potential for miniaturization but also significantly enhances joint quality in micro-welding applications.

The core advantage of this system lies in its high-frequency (1.76 MHz) and miniaturized design, enabling high-precision energy control. This makes it particularly suitable for applications sensitive to thermal input (e.g., microelectronic packaging, precision medical devices), which is urgently needed for welding many composite and dissimilar materials. Accordingly, we expanded our investigation to assess the feasibility of welding different materials. Experiments revealed successful welding of dissimilar materials such as PVC to metal plates (Figure 18a); PVC to polyethylene sheets (PE) (Figure 18c); PVC and paper (Figure 18d). Within the same material category, Polymethyl methacrylate sheet (PMMA) used as a microfluidic chip (Figure 18b) achieved satisfactory weldability. The successful welding of these materials validates the effectiveness of this research methodology, laying a solid foundation for subsequent studies on more complex material systems.

## 5. Conclusions

Based on the theory of acoustic propagation and transmission, this study designed a cylindrical miniature ultrasonic welding oscillator operating at the terahertz level. Finite element simulation results showed strong agreement with experimental measurements, confirming both the accuracy of the modeling approach and the effectiveness of the structural design. Modal and harmonic response analyses identified a resonant frequency of approximately 1.76 MHz, which aligns closely with the target operating frequency of the piezoelectric ceramic element.

Laser Doppler vibrometry measurements demonstrated effective energy localization, with the working surface amplitude exceeding 50 nm, while vibrations on non-functional surfaces were consistently below 5 nm, indicating minimal parasitic loss. The deviation between simulated and experimental amplitude distributions on the working surface was less than 5%, underscoring the predictive reliability of the finite element model. This high level of agreement is essential for reducing prototype iterations and accelerating the development of high-frequency micro-ultrasonic systems. These results represent a significant advancement in terahertz-level micro-ultrasonic welding, achieving precise amplitude control and spatial energy confinement within a compact form factor (Φ28 mm × 18 mm).

Welding tests on PVC materials using the prototype system revealed that optimal weld strength—exceeding 45 N in tensile force—was achieved under the following parameters: 85 N welding pressure, 4 s welding time, and 3 s holding time. Compared to conventional low-frequency ultrasonic welding systems, the proposed design reduces energy consumption by approximately 37%.

This study provides not only a miniaturized and optimized hardware solution but also experimentally validated process guidelines for high-frequency micro-ultrasonic welding. These contributions support the adoption of precision joining technology in space-constrained applications such as microelectronics and medical devices, while promoting energy-efficient and sustainable manufacturing practices.

## Figures and Tables

**Figure 1 sensors-25-05922-f001:**
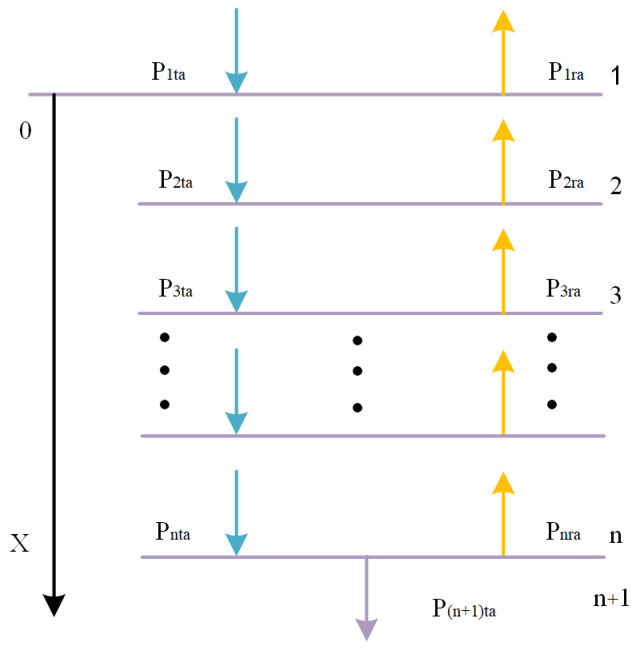
Acoustic transmittance model.

**Figure 2 sensors-25-05922-f002:**
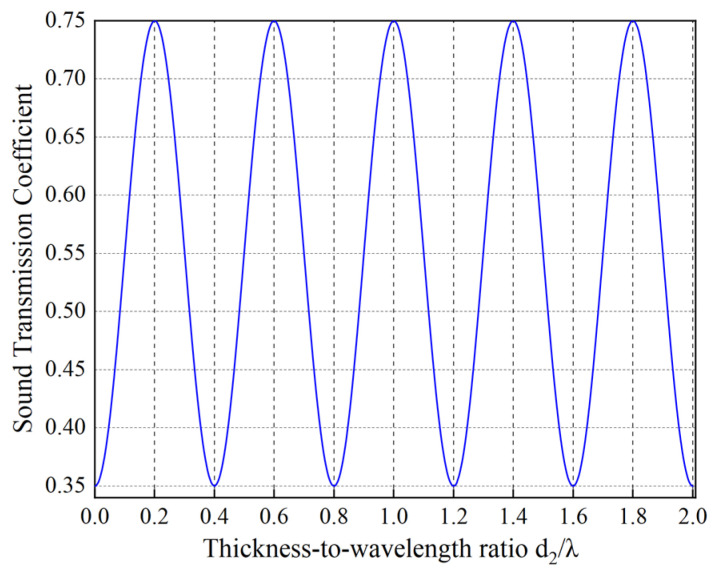
Acoustic projection Curve of a 2 MHz frequency transducer.

**Figure 3 sensors-25-05922-f003:**
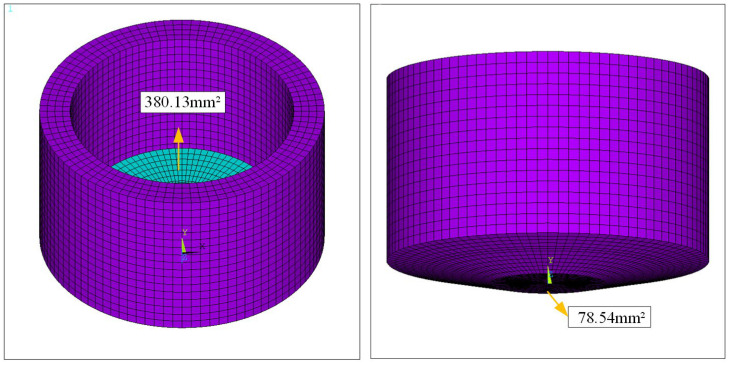
Parametric modeling of ultrasonic welding oscillators.

**Figure 4 sensors-25-05922-f004:**
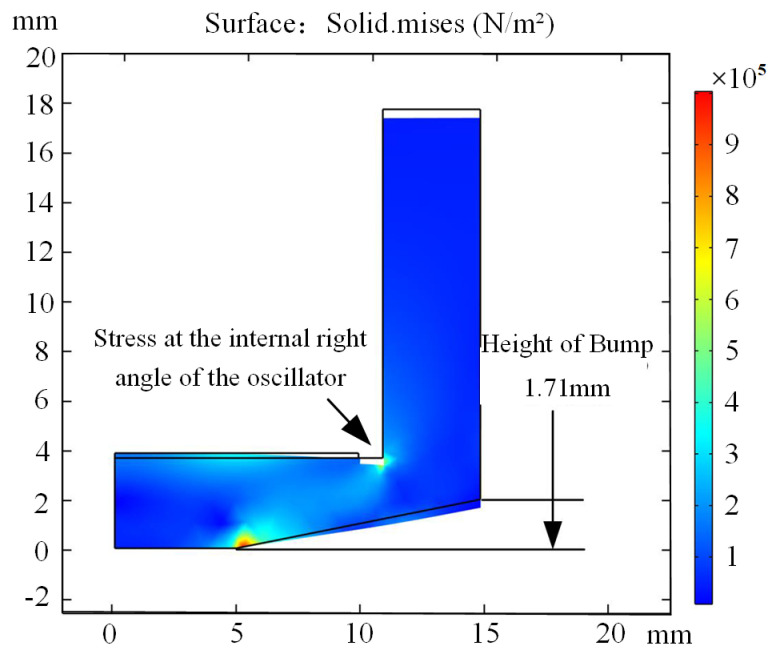
Stress analysis cloud image of ultrasonic welding oscillator.

**Figure 5 sensors-25-05922-f005:**
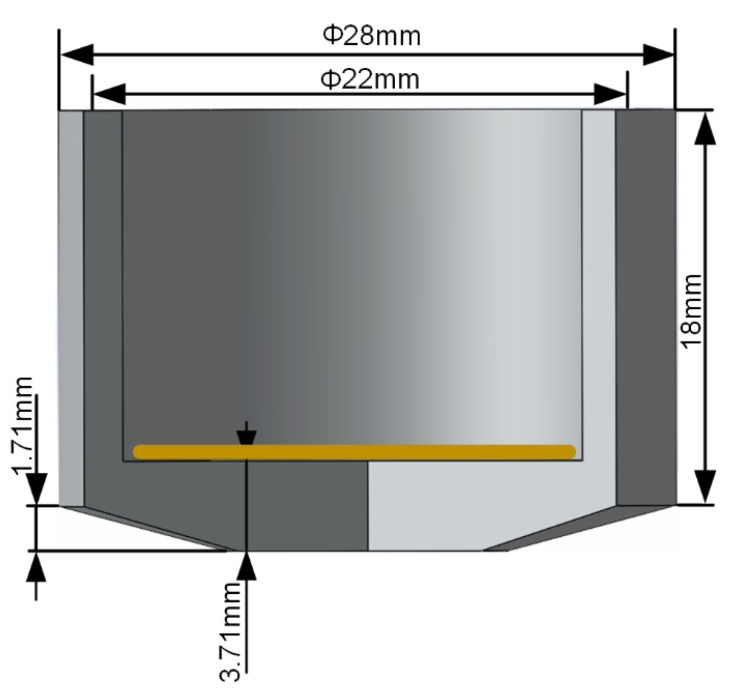
Ultrasonic welding oscillator structure holding dimensions.

**Figure 6 sensors-25-05922-f006:**
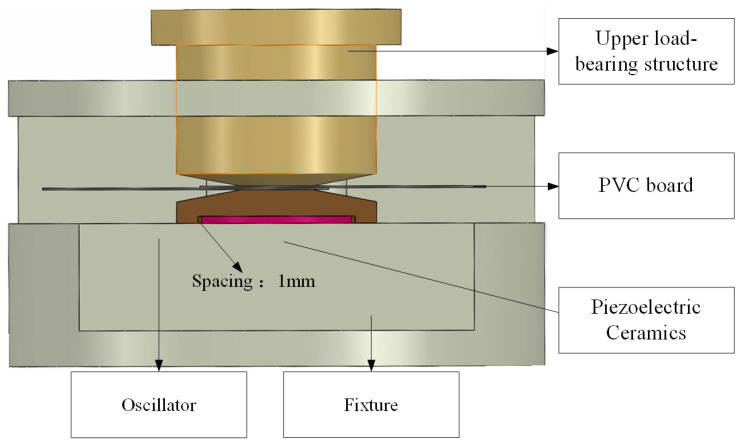
Welding System component diagram.

**Figure 7 sensors-25-05922-f007:**
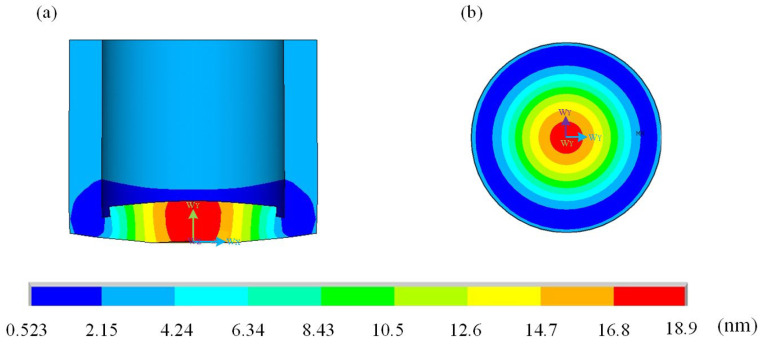
Displacement cloud image of micro welding oscillator. (**a**) Displacement cloud of micro-welding oscillator profile. (**b**) Top-view displacement cloud of a miniature welding oscillator.

**Figure 8 sensors-25-05922-f008:**
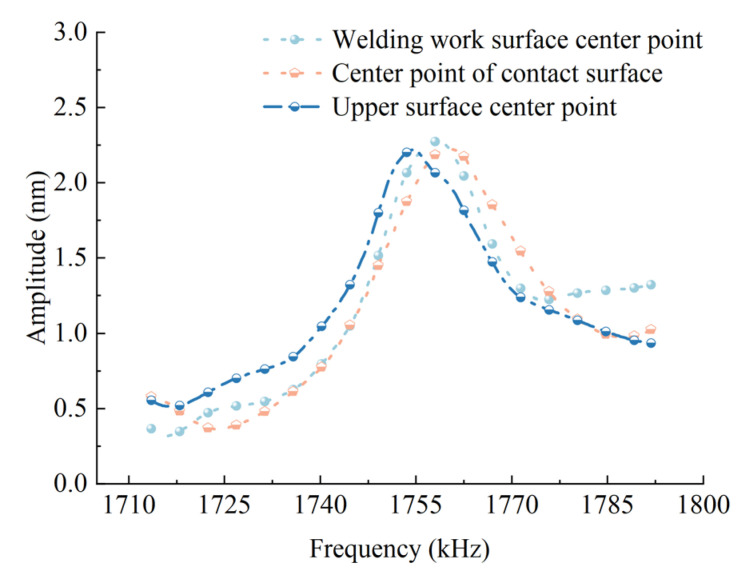
Ultrasonic welding oscillator frequency response curve.

**Figure 9 sensors-25-05922-f009:**
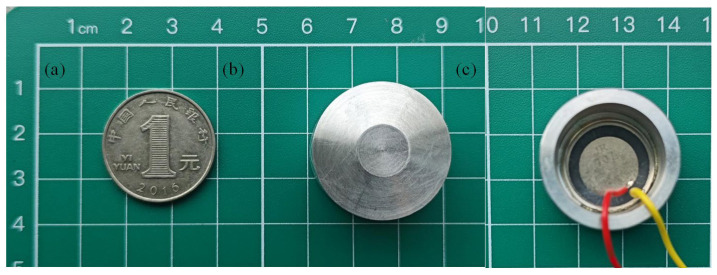
Miniature ultrasonic welding oscillator. (**a**) General-purpose coin made in China: 25 mm diameter; (**b**) Processing of ultrasonic welding oscillator; (**c**) oscillator assembly with pasted piezoelectric ceramic sheet.

**Figure 10 sensors-25-05922-f010:**
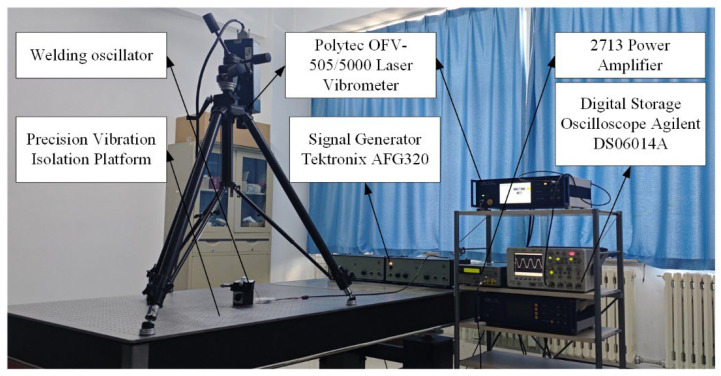
Test system of oscillator.

**Figure 11 sensors-25-05922-f011:**
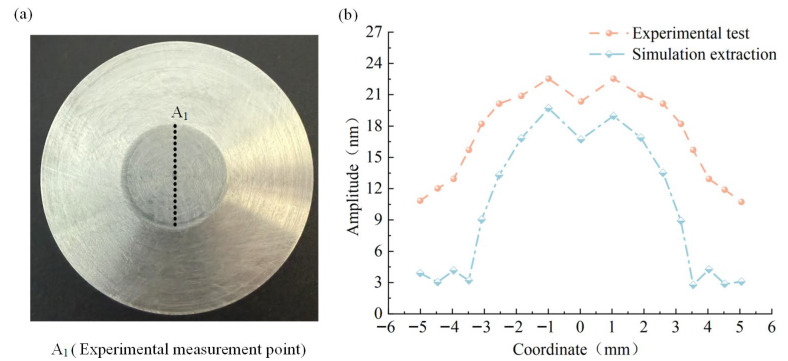
Amplitude map of test points in the working face. (**a**), Testing points on the working face of the welding oscillator; (**b**), The amplitude distribution of welding oscillator’s working surface.

**Figure 12 sensors-25-05922-f012:**
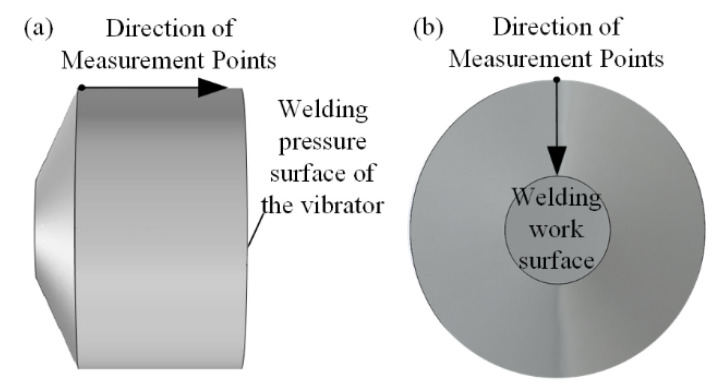
Non-Working face amplitude test scheme of welding oscillator. (**a**), Vibrating Wall Test Program; (**b**), Bump Bevel Test Program.

**Figure 13 sensors-25-05922-f013:**
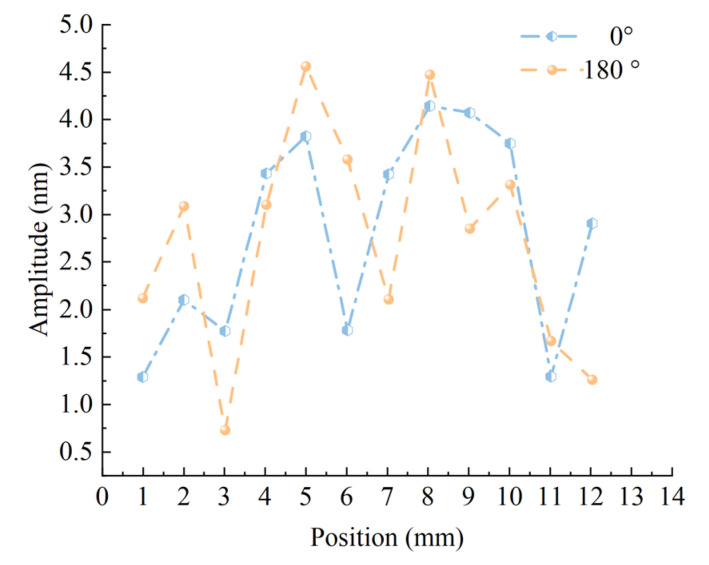
Amplitude distribution of welded oscillator wall.

**Figure 14 sensors-25-05922-f014:**
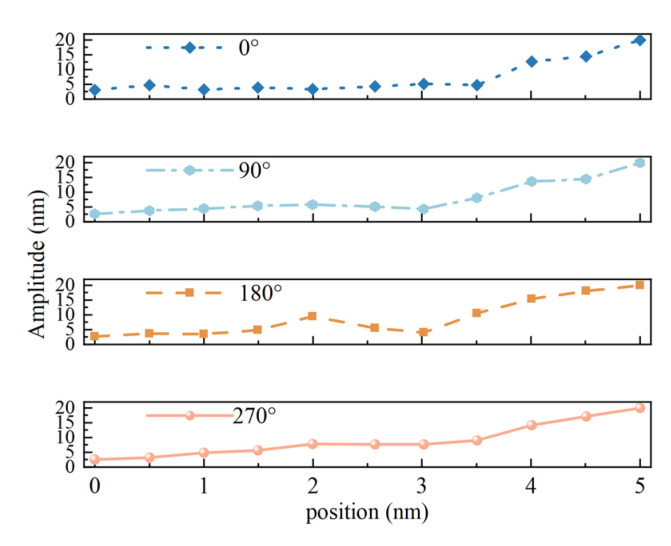
Amplitude distribution of the bevel of welding oscillator boss height.

**Figure 15 sensors-25-05922-f015:**
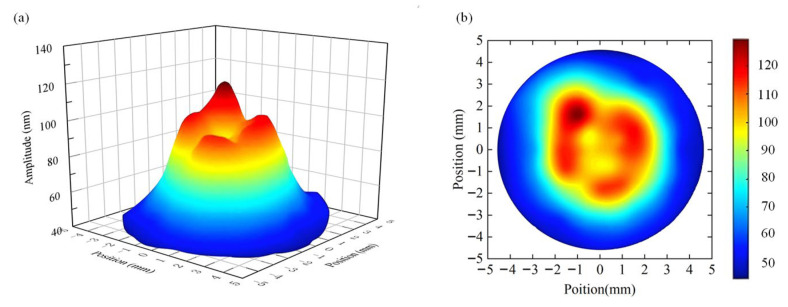
Amplitude test results. (**a**), Amplitude Distribution Cloud Map of Welding Work Surface; (**b**), Top view of the amplitude distribution of the welding work surface.

**Figure 16 sensors-25-05922-f016:**
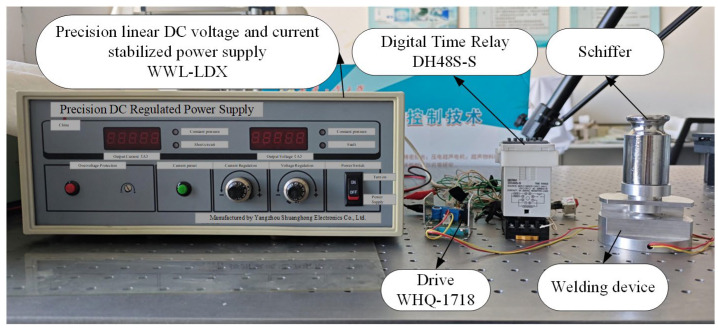
Micro ultrasonic welding System.

**Figure 17 sensors-25-05922-f017:**
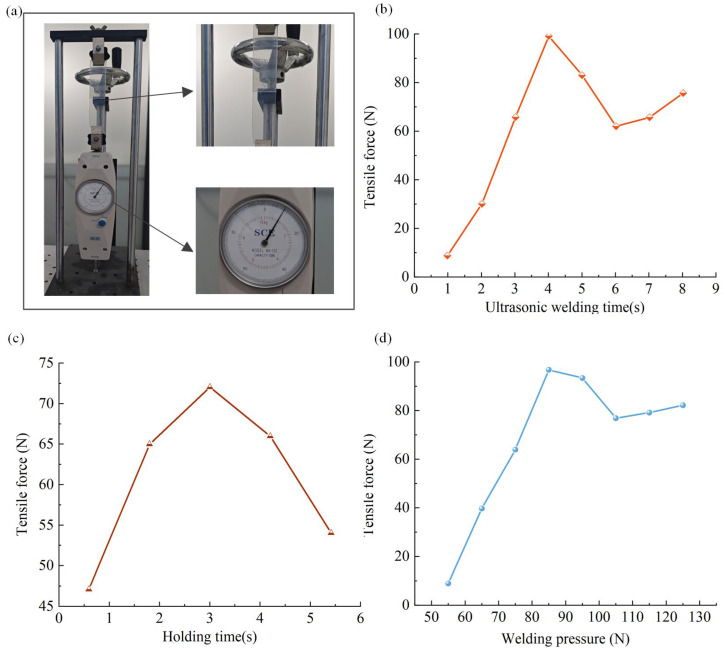
Process Parameter Influence Diagram. (**a**), Experimental tension test system; (**b**), Relationship between pressure holding time and tensile force; (**c**), Relationship between welding time and tensile force; (**d**), Relationship between welding pressure and tensile force.

**Figure 18 sensors-25-05922-f018:**
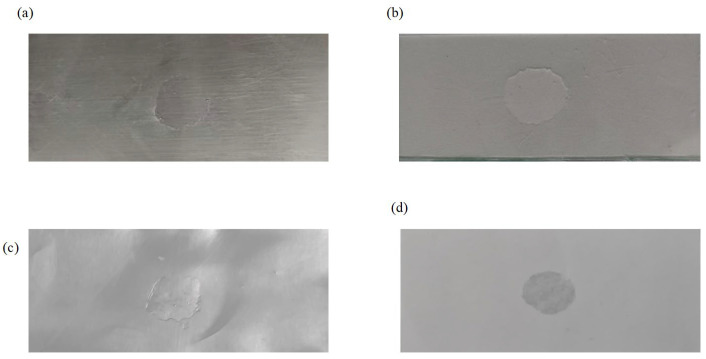
Multi-Material Welding Diagram. (**a**), PVC and metal sheet. (**b**), Polymethyl methacrylate sheet (PMMA); (**c**), Polyethylene Sheet (PE); (**d**), PVC and paper.

**Table 1 sensors-25-05922-t001:** Welding oscillator summary.

System	Frequency	Scale	Approx. Energy Per Weld	Key Feature/Limitation
Conventional System	28 kHz	Φ100 × 200	150 J	Large size, high power, suitable for macro-parts.
Parrini et al. [20]	125 kHz			High frequency but long joining time.
SAW System	1.2 MHz			High frequency, zero radial displacement.
This Work	1.76 MHz	Φ28 × 18	65.8 J	Miniaturized, energy-efficient, precise for micro-welding.

**Table 2 sensors-25-05922-t002:** Material parameter table.

Medium	Wave Number (m^−1^)	Acoustic Impedance Value (10^6^ N·s/m^3^)
PZT-4	2670	32.072
Aluminum	1690	18.495
PVC	5341	3.343

**Table 3 sensors-25-05922-t003:** Matching layer thickness of different wavelengths.

Wavelength	Matching Layer Thickness (mm)
λ/5,(n=0)	0.74
3λ/5,(n=1)	2.23
λ,(n=2)	3.71
7λ/5,(n=3)	5.19

**Table 4 sensors-25-05922-t004:** Parameters of piezoelectric ceramic components.

Piezoelectric Ceramic Components	Parameters
Dimensions	Diameter Φ: 20 mm; Electrode face diameter Φ: 13 mm; Thickness T: 1.2 mm;
Materials	Oscillator density: ρ = 7559 kg/m^2^; Poisson ratio: σ = 0.32
Spring constant	C_11_ = 13.9, C_12_ = 7.78 C_13_ = 7.43, C_33_ = 11.5, C_44_ = 2.56, C_66_ = 3.06
Piezoelectric stress	e_15_ = 12.7, e_31_ = −5.2, e_33_ = −15.1
Dielectric constant	ε_11_ᵀ = 3.276, ε_33_ = 5.622
Young’s modulus	Anisotropic

**Table 5 sensors-25-05922-t005:** Finite element simulation parameter table.

Young’s Modulus	72 GPa	Modal Analysis Solver	Block Lanczos Solver
Poisson ratio	0.33	Harmonic Response Analysis	Mode Superposition
Density	2700 kg/m^3^	Unit Type	SOLID45
Lower cylindrical end face	Fixed Constraint	Unit Size	Maximum dimension ≤ λ/10 ≈ 0.36 mm
Excitation voltage	48 V	Grid Convergence	Ensure amplitude variation remains below 2% after mesh refinement.
Grounding electrode	lower surface	Structural Damping Coefficient	0.005
stimulation electrode	Upper surface		

## Data Availability

The data that support the findings of this study are available from the corresponding author upon reasonable request.

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
