# Peer review of "Design of a Cylindrical Megahertz Miniature Ultrasonic Welding Oscillator"

_sensors, 2025, doi:10.3390/s25185922_

Round 1
Reviewer 1 Report
Comments and Suggestions for Authors
This paper attempts to design a cylindrical micro-welding oscillator with a diameter of 28 mm, a height of 18 mm, and an operating frequency of approximately 1.76 MHz. The authors claim that the optimal matching layer thickness is 3.71 mm, and that a peak tensile force of approximately 45 N can be achieved on a PVC specimen. The optimal parameter combination is: welding pressure of 85 N, welding time of 4 s, and holding pressure of 3 s. The paper also provides schematic frequency response and working surface amplitude measurements, but there is currently a gap between theoretical derivation and empirical design, making it difficult to verify novelty and reproducibility.
- The relevant literature is well-written, but several important works are missing. I have listed some relevant studies for the authors to reference in evaluating their research results. This will provide readers with a clearer understanding of the missing content in the literature and how this paper addresses these issues.
(1) Huang, S. H., Wu, T. E., & Lai, C. H. (2025). Automated Detection of Arc Welding Defects by Using a Convolutional Neural Network Model. Journal of the Chinese Society of Mechanical Engineers, 46(2), 205-210.
(2) Dong, Z., & Shao, C. (2025). Fine-Scale Characterization and Monitoring of Tool Surface Degradation in Ultrasonic Metal Welding Using Optical Measurements and Computer Vision. Journal of Computing and Information Science in Engineering, 1-19.
- The authors emphasize the value of MHz-level ultrasonic welding in space-constrained applications, but the description of the bottlenecks of existing low-frequency technologies is still a bit vague. The quantitative differences in insufficient energy transmission efficiency or volume limitations should be more specifically explained.
- The derivation of the acoustic transmission theory is complete, but lacks assumptions. The derivation does not clearly state which energy losses are ignored.
- Only test data from a single specimen is presented, lacking repeated experiments and statistical analysis, which limits the generalizability of the conclusions.
- Welding tests were conducted only on PVC, and the feasibility of welding composite materials or metal/plastic dissimilar materials was not demonstrated.
- The authors present the thickness sequence using Equation (20) and Table 2, and select n = 2 (3.71 mm) as the final design, but do not present parameter sweeps or objective optimization criteria for the transmission coefficient/sound intensity; nor do they compare the results with the physical intuition of matching a typical quarter-wavelength. Please clearly define the objective function.
- Only a subset of material parameters are presented, and the piezoelectric coupling constant, dielectric constant, boundary conditions, electrode/adhesive layer modeling, damping, and mesh convergence are not listed. A complete FE parameter table and setting values should be provided to increase the repeatability of the experiment. 8. What does "all opposite sex" in Table 3 mean?
- While the welding experimental design describes a single-factor test to control welding pressure/welding time/holding pressure, the experimental design (parameters such as specimen size, overlap area, thickness, number of samples, and error bars) is not explained. A clear definition is recommended.
- The conclusion claims a 37% reduction in energy consumption compared to conventional systems, but the measurement methods and statistical evidence are not provided throughout the paper. This should be clarified.
- The introduction mentions various welding systems, but does not quantify the differences in size, energy consumption, strength, or cycle time between this study and these systems. A comparison table should be added to highlight the novelty of this study.
By completing the theoretical-simulation-experimental linkage and refining the methodology and statistics, this paper has the potential to become a reproducible design example for MHz-class plastic welding actuators. At this stage, it is recommended that the above issues be revised; otherwise, the paper will not be accepted.
Reviewer 2 Report
Comments and Suggestions for Authors
Dear Authors,
Your manuscript discusses the ultrasonic welding oscillator with high frequency (MHz). In general, the manuscript is clearly written and well-structured. However, there are several issues that need clarification and improvement:
- Introduction: You referred to many studies in this field. However, for the MHz frequency, please provide more background information and make the motivation for conducting this research clearer.
- Line 178–182: Please explain the reason for selecting these parameter values. Are they based on equipment limitations or other considerations?
- Simulation model (Figure 2):
- In Figure 2, please indicate the area of the piezoelectric ceramic and the oscillator.
- Please provide more explanation for selecting the “Solid45” element (Table 3).
- Include details about the meshing (mesh size, meshing method, etc.).
- Please specify the boundary conditions of both the simulation model and the experimental model.
- For both the simulation and experimental models, please provide a figure showing the dimensions and the assembly between the PZT and the oscillator.
- Figure 3: Please indicate the location of the section shown.
- Figure 4: Please add the PZT ceramic location or clarify the contact between the oscillator and the PZT ceramic.
- The quality of all result figures is not clear. Please improve the resolution and clarity.
- Results: For the comparisons between simulation and experiment, please provide more detailed descriptions and explanations of the results. If these results are novel, please highlight their significance and potential applications.
- Tensile testing: Do you have any comparisons between your results and those of other studies, especially considering the high frequency you used?
Sincerely yours,
Round 2
Reviewer 1 Report
Comments and Suggestions for Authors
Thank you for carefully addressing the previous review comments. The revised manuscript has adequately responded to the suggestions, with improved clarity and rigor in its presentation. I consider the manuscript to have reached the publication standard of this journal.
Reviewer 2 Report
Comments and Suggestions for Authors
Dear Authors,
In general, the revised version of your manuscript is good. However, there are still some minor issues that should be addressed:
- Figure 5 (Welding System Component Diagram): Please consider adding color or another method to make it easier to distinguish the different parts.
- The quality of the figures has been improved; however, if possible, please make them clearer.
Sincerely yours,
